# Machine Learning-Based Void Percentage Analysis of Components Fabricated with the Low-Cost Metal Material Extrusion Process

**DOI:** 10.3390/ma15124292

**Published:** 2022-06-17

**Authors:** Zhicheng Zhang, Ismail Fidan

**Affiliations:** 1Department of Mechanical Engineering, Tennessee Tech University, Cookeville, TN 38505, USA; zzhang44@tntech.edu; 2Department of Manufacturing and Engineering Technology, Tennessee Tech University, Cookeville, TN 38505, USA

**Keywords:** additive manufacturing (AM), low-cost metal material extrusion (LCMME), sintering, microstructure, machine learning (ML)

## Abstract

Additive manufacturing (AM) is a widely used layer-by-layer manufacturing process. Material extrusion (ME) is one of the most popular AM techniques. Lately, low-cost metal material extrusion (LCMME) technology is developed to perform metal ME to produce metallic parts with the ME technology. This technique is used to fabricate metallic parts after sintering the metal infused additively manufactured parts. Both AM and sintering process parameters will affect the quality of the final parts. It is evident that the sintered parts do not have the same mechanical properties as the pure metal parts fabricated by the traditional manufacturing processes. In this research, several machine learning algorithms are used to predict the size of the internal voids of the final parts based on the collected data. Additionally, the results show that the neural network (NN) is more accurate than the support vector regression (SVR) on prediction.

## 1. Introduction

Additive manufacturing (AM), also known as 3D printing (3DP) [1], is a set of technologies that produce objects layer by layer from the computer-aided design (CAD) model [2]. At the present stage, there are seven different AM categories, which are material extrusion (ME), vat photopolymerization (VAT), powder bed fusion (PBF), direct energy deposition (DED), sheet lamination (SL), material jetting (MJ), and binder jetting (BJ) [3]. Among these techniques, ME is well known and the most widely used [1] because of its numerous advantages such as producing complex parts with less materials and time [4], low-cost [4], environmental friendliness [5], etc. The ME technique is popular in several research areas, such as the food industry [6], medicine [7], aerospace [8], and so on. In recent years, ME has been used in metal object manufacturing [9]. However, ME is not widely used in metal 3DP because the working temperature of ME 3D printers is much lower than the melting temperature of most metals [1]. Thus, low-cost metal material extrusion (LCMME) is developed to use metal-infused polymer filaments and fabricate metal components with a low-cost manufacturing process [10].

Sintering is a manufacturing process used in the post processing operations. It is used to form a solid mass of metal by heating the fabricated parts at a temperature that is just a little lower than the melting point of the metal [11]. In the LCMME process, the metal–infused polymer filament is melted and extruded with the help of a 3D printer to produce the specimens. Then, the parts are sintered to extract the metal part out of the metal infused polymer part by melting the polymer material [12]. The sintering process is crucial because the mechanical properties of metal–polymer composite parts are much lower than pure metal [9]. By heating the composite parts to just below the melting point of the metal, the internal polymer melts and evaporates, which will lead to a pure metal part eventually. However, in Mohammadizadeh’s research, there were some voids inside the final parts [9]. Thus, the most suitable manufacturing parameters are required to investigate to minimize the size of internal voids.

The microstructure of AM fabricated parts has been studied in some research and different part microstructure will lead to different mechanical properties. In the study of Mohammadizadeh et al., if the microstructure of LCMME parts is different, the tensile test result will be different [9]. Additionally, in their research, Mohammadizadeh et al. pointed out that the final LCMME-fabricated parts will have some internal voids, which are shown in Figure 1.

Gubicza et al. studied the correlation between the microstructure and the mechanical behavior of metals [13]. Reiser et al. found that in the AM process, the microstructure will influence the mechanical properties [14]. The large variation in materials’ performance can be related to the individual microstructure. In several studies, the materials used in AM are different, but all the results show that the microstructure and mechanical properties have a strong relationship [15,16,17].

Machine learning (ML) is a subset of artificial intelligence and it is widely used to predict the mechanical properties of fabricated parts [18]. ML is the process to use computer algorithm, regression, or classification models to make predictions on the data without using any exact methods [1]. It has the ability to self-train from the given data and makes the decision by itself [19]. ML has been used to help users in various areas, such as dimensional accuracy prediction of LCMME-fabricated parts [1], language detection and translation [20], medicine [21], etc.

ML has been widely used in AM, and there are plenty of studies on microstructure. However, there is no study on using ML to predict and improve the microstructure of LCMME-fabricated parts. In this study, the main aim is to improve the quality of the parts fabricated by LCMME with the help of ML. Firstly, cuboid samples were fabricated by the LCMME process, and the influence of different manufacturing parameters on internal voids after sintering was analyzed. Then, two different ML algorithms were generated to make predictions on the microstructure quality.

## 2. Materials and Methods

### 2.1. Materials and Equipment

The metal–polymer composite material used in this research is the bronze–PLA filament made by The Virtual Foundry (Stoughton, WI, USA) [22]. The parts were fabricated in an Ultimaker S5 3D printer, which is made by the Ultimaker (Utrecht, The Netherlands) [23]. Then, the 3D printed parts were sintered in a KSL-1100X muffle furnace, made by the MIT Corporation (Richmond, CA, USA) [24]. A 35-025 electronic micrometer, made by iGaging (San Clemente, CA, USA) [25], was used to measure the sample dimensions. Additionally, the mass of the sintered samples was measured by a US-KA6 AMIR Pocket Scale which is made by AMIR (Karachi, Pakistan) [26]. 

The material and equipment used in this research are shown in Figure 2. Figure 3 shows the metal-composite part as a 1. CAD model, 2. 3D printed part, and 3. sintered final component. The solid models of the test parts are sliced with the use of Cura 4.13.1 software, which is developed by Ultimaker. Using Ultimaker S5 printer and Cura slicer help the research team produce very accurate test parts.

### 2.2. Process Workflow

The schematic of this research is shown in Figure 4. There are three main sections in this research. The first section is Data Collection. The G-code was generated from a CAD model in Cura, then sent it to Ultimaker S5 to fabricate the bronze–PLA parts. After the 3DP process, the samples were sintered in the KSL-1100X muffle furnace. Then, the dimensions and mass of sintered parts were measured. The second section is Data Preparation. The internal void percentages of samples were calculated, the calculation process will be introduced in Section 2.4. Then, by using ANOVA, the influence of each manufacturing parameters was analyzed. The last section is ML prediction. Two different ML algorithms were developed to help the users predict the microstructure quality before the LCMME process, and the accuracy of these two ML algorithms were compared.

### 2.3. Sintering Process Introduction

Figure 5 is a sketch of the sintering process. As the temperature increases, the plastic melts out and the metal powder gathers together to form pure metal parts. However, in the sintering process, since the sintering temperature is lower than the melting temperature [11], an internal void cannot be avoided in the final part. However, different manufacturing parameters will lead to different sizes of voids.

The material used in this research is a bronze–PLA composite filament. Figure 6 shows two micro view photos of non-sintered material (a) and the final pure bronze part (b).

### 2.4. Parameter Introduction and Data Collection

In this research, there are five different manufacturing parameters, which are:**Layer Thickness (LT):** the height of each layer during the printing process;**Sintering Temperature (ST):** the temperature to sinter the bronze–PLA parts;**Ramp Ratio (RR):** the temperature increasing ratio from room temperature to ST;**Nozzle Temperature (NT):** the temperature of the printing nozzle during the 3DP process;**Printing Speed (PS):** the moving speed of the nozzle during the 3DP process.

Table 1 shows the units and values of different manufacturing parameters.

In this research, 150 samples were fabricated and resulted in 450 groups of data points. The length (L), width (W), height (H), and mass (M) were measured. Since the samples in this research are cubes, the volumes (V) of the samples can be easily calculated by: V = L × W × H(1)

Then, the volume is multiplied by the density of bronze (ρ), and a calculated mass (M_Cal) is obtained:M_Cal = V × ρ(2)

If there were no internal voids, M would be the same as M_Cal. Since there are internal voids in the final parts, M and M_Cal will be different. Additionally, the different percentage between M and M_Cal will be the void percentage:Void percentage = (1 − M/M_Cal) × 100%(3)

Table 2 shows two examples of void percentages.

The first example was fabricated with 0.1 mm LT, 875 °C ST, 3 °C/min RR, 220 °C NT, and 10 mm/s PS; the real mass is 7.25 g, but the calculated mass is 8.15 g. So, the void percentage of the first example is 12%. Additionally, the second had different manufacturing parameters, and the void percentage is 48%. Compared with the void percentages of the above two examples, the first sample has a smaller void percentage than the second, which means that this sample is more solid and the mechanical property will be better [9], and thus, the quality of the first one is better than the second one.

### 2.5. Introduction of ML Algorithms

In this research, two different ML algorithms were developed, such as the support vector regression (SVR) and neural network (NN). The following subsections will explain the definitions and reasons for choosing these two kinds of algorithms.

#### 2.5.1. SVR

SVR is a type of supervised ML algorithm and it is a unique linear regression method. It is used in this research because it has been proven to be an effective tool in real-value function estimation [27]. SVR could provide flexibility to define the error in the model is acceptable or not. In 2D application, it will give an appropriate line to fit the data [28]. The constraint of SVR is:|yi − wixi| ≤ ε(4)
where yi is the real dependent variable, wi is the coefficient, xi is the independent variable, and ε is the acceptable error [28]. 

#### 2.5.2. NN

NN is a kind of widely used ML algorithm which could be used both in supervised and unsupervised learning [29]. It has a set of network layers to translate the input data to output [30]. NN uses multiple layers of linear processing units for feature extraction and transformation. In each layer, the input is the output of the previous layer. In this research, a supervised manner is used since the dependent variables are labeled. Additionally, there is more than one hidden layer, so a deep NN model is developed. 

The schematic of the NN is represented in Figure 7. Input the manufacturing parameters and void percentages in the input layer, the data is analyzed in these hidden layers, and then the NN algorithm could be used to predict void percentage and output from the output layer.

## 3. Results

In this section, several plots are used to show the influences of manufacturing parameters on void parameters and the results of two ML algorithms are presented. 

### 3.1. Influence of Manufacturing Parameters

There are five different manufacturing parameters analyzed in this research, such as LT, ST, RR, NT, and PS. Table 3 shows the ANOVA analysis, wherein the p-values of all manufacturing parameters are smaller than 0.001. Thus, all manufacturing parameters have significant influence on void percentage. Two-way and three-way interactions are also observed in the ANOVA analysis and also creating significant impact on the void percentage.

Figure 8 is the boxplot showing the change in void percentage with the change in each manufacturing parameter. In the different five boxplots, the horizontal axis shows the manufacturing parameters and the vertical axis is always the internal void percentage. As LT and PS increased, the void percentage will increase. Additionally, with RR increasing, void percentage will decrease. The void percentage will increase than decrease as NT increased from 220 °C to 240 °C. As ST increased from 870 °C to 900 °C, the trend of void percentage is decreasing firstly then increasing.

### 3.2. Interactions of Manufacturing Parameters

From the results of the ANOVA analysis (Figure 8) above, there are several interactions between the manufacturing parameters. The research group used a microscope to collect the micro view photos of the samples. In Figure 9, the top sample was fabricated with the following parameters: LT: 0.1 mm, ST: 870 °C, RR: 4 °C/min, NT: 220 °C, and PS: 10 mm/s. The bottom has the manufacturing parameters: LT: 0.2 mm, ST: 880 °C, RR: 3 °C/min, NT: 220 °C, and PS: 10 mm/s. From these two samples, the size of the internal voids is significantly different, the void in bottom one increases obviously, because the manufacturing parameters are different. Thus, by changing the manufacturing parameters, the size of the internal voids could be minimized.

### 3.3. Comparison of Different ML Algorithms

In the study of Zhang et al. [1], they use the same manufacturing method, LCMME, and four different ML algorithms are used to predict the dimensional accuracy of this unique manufacturing process. Their results show that simple linear regression and linear regression with interaction behave worse than NN and SVR. Thus, in this research, these two better algorithms are used, not four.

Since in ANOVA, all parameters will have influence on the internal void percentage of the additively manufactured parts, in the prediction process, the independent variable matrix is:x_i_ = [LT, ST, RR, PS, NT](5)

Additionally, the error vector is fixed:ε = [0.1, 0.1, 0.1, ……](6)

The structure of NN has been shown in Figure 7. In this research study, it has five hidden layers, and there are 128 neurons in each hidden layer. The activation function is ReLU.

To compare the behaviors of these two ML algorithms, the mean square errors (MSE) are calculated. 

MSE is the average squared difference between the estimated values (predicted values) and the actual value (observed values) [31]. Additionally, smaller MSE leads to a more accurate algorithm [32]. The equation is given below:(7)$$MSE=\frac{1}{n}\overset{n}{\underset{i=1}{\sum }}{\left(Y_{i}-{\hat{Y}}_{i}\right)}^{2}$$
where MSE is the mean square error, n is the sample size used to test an algorithm, Yi is the observed value, and Ŷi is the value predicted by the algorithm. 

Table 4 shows the MSE values of the two ML algorithms. Obviously, NN has a smaller MSE value than SVR. Thus, NN behaves better than SVR in microstructure prediction. Additionally, the microstructure data collected in this research varies from 0.12 to 0.55, and the MSEs are quite small compared with the data. So, these two ML algorithms have high accuracy in predicting microstructure of LCMME-fabricated parts.

## 4. Discussions

This research study investigated the relationships between the LCMME process parameters and the internal void percentage of the final parts. Based on the findings of the performed studies, the influence of each process parameter on microstructure has been presented. Additionally, two ML algorithms were used to predict the void formation.

In the LCMME process, it is evident that while the values of the control parameters change, the microstructure will change. If we want to have a very minimal internal void, the suggested manufacturing parameters will be as follows:LT: 0.3 mmST: 895 °CNT: 240 °CPS: 20 mm/sRR: 4 °C/min

For real-world/industrial applications, more mechanical and physical properties of the final parts need to be considered in the future, such as surface roughness, density, dimensional accuracy, etc. Adding such parameters might cause small fluctuations, compared to the results of this research study, in establishing the relationships between the process parameters and internal voids.

The current results of the ML-based void percentage analysis help the AM practitioners predict the mechanical and physical properties before LCMME process, and then the parameters generated from the ML prediction will be employed so that the quality of the additively manufactured parts is reliable.

## 5. Conclusions

In the area of metal AM, there is strong correlation between microstructure and mechanical properties of the final parts. In order to improve the quality of LCMME-fabricated parts, by using ML and analyzing the size of the internal void in the LCMME process, the following conclusions are found:In this research, all manufacturing parameters have influence on the void percentage;Different manufacturing parameters have different influences on the final void size;These manufacturing parameters have interactions among them, changing two parameters will cause the void size to vary significantly;ML algorithms could be used to predict the void percentage before the manufacturing process, and the accuracies of both NN and SVR are quite high;NN provides more accurate results than SVR.

## 6. Future Work

This work used a statistical method to prove the influence of manufacturing parameters on the microstructure quality of LCMME-fabricated parts. Additionally, two different ML algorithms are developed to predict the percentage of the void. In the future, manufacturing parameters such as flow rate, build-plate temperature, fan speed, etc., could be added to decrease the size of the void. In addition, the algorithms could be improved to be more accurate.

## Figures and Tables

**Figure 1 materials-15-04292-f001:**
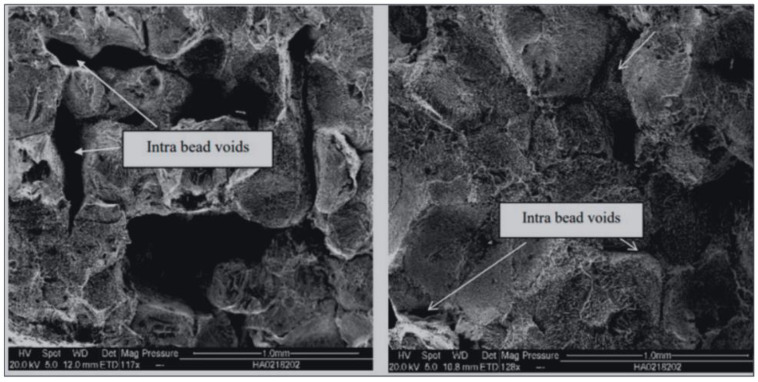
SEM image of LCMME-fabricated copper part [9].

**Figure 2 materials-15-04292-f002:**
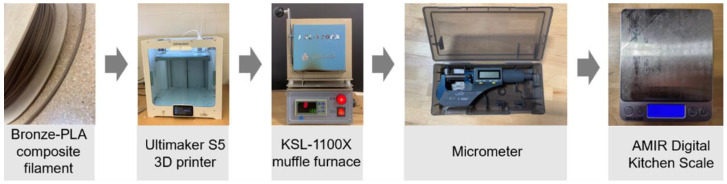
Material and equipment used in this research.

**Figure 3 materials-15-04292-f003:**
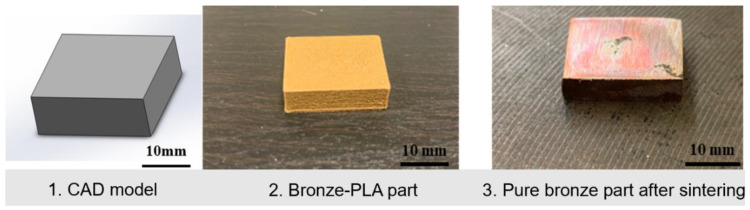
Samples in different status.

**Figure 4 materials-15-04292-f004:**
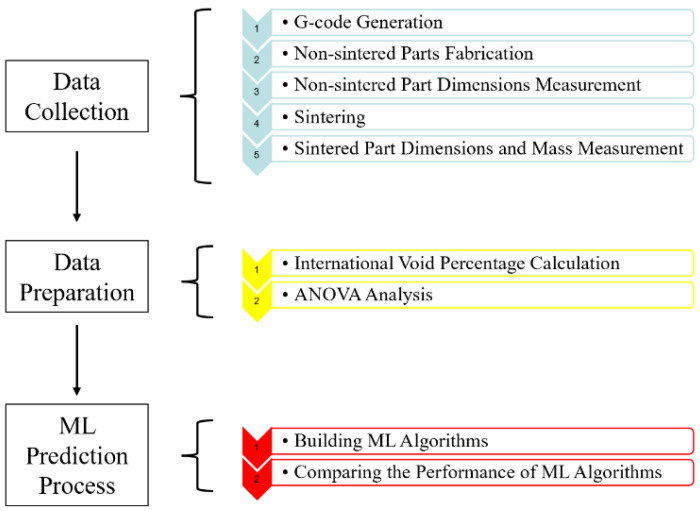
Process workflow of the research.

**Figure 5 materials-15-04292-f005:**
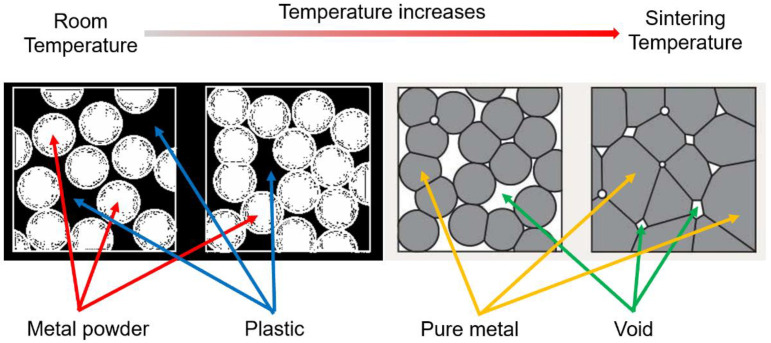
Sketch of sintering process.

**Figure 6 materials-15-04292-f006:**
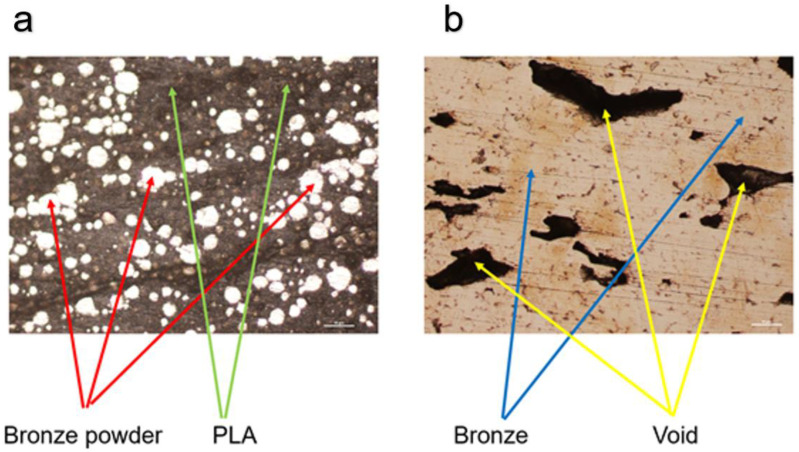
Micro view photos of non-sintered material (**a**) and final pure bronze part (**b**).

**Figure 7 materials-15-04292-f007:**
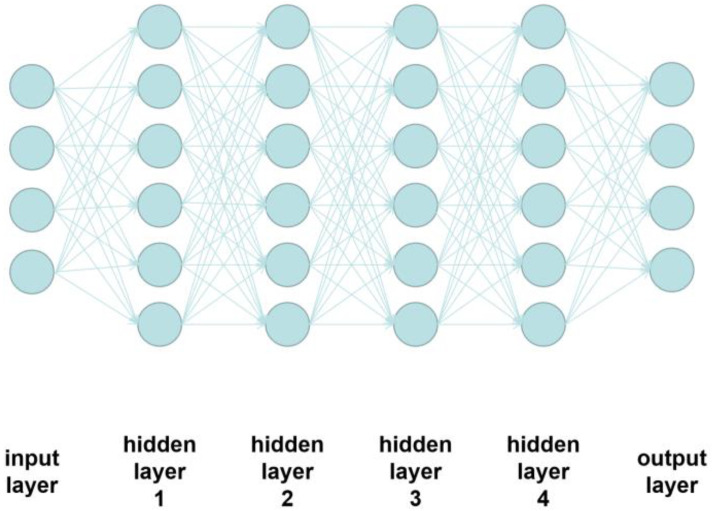
Schematic of the NN [5].

**Figure 8 materials-15-04292-f008:**
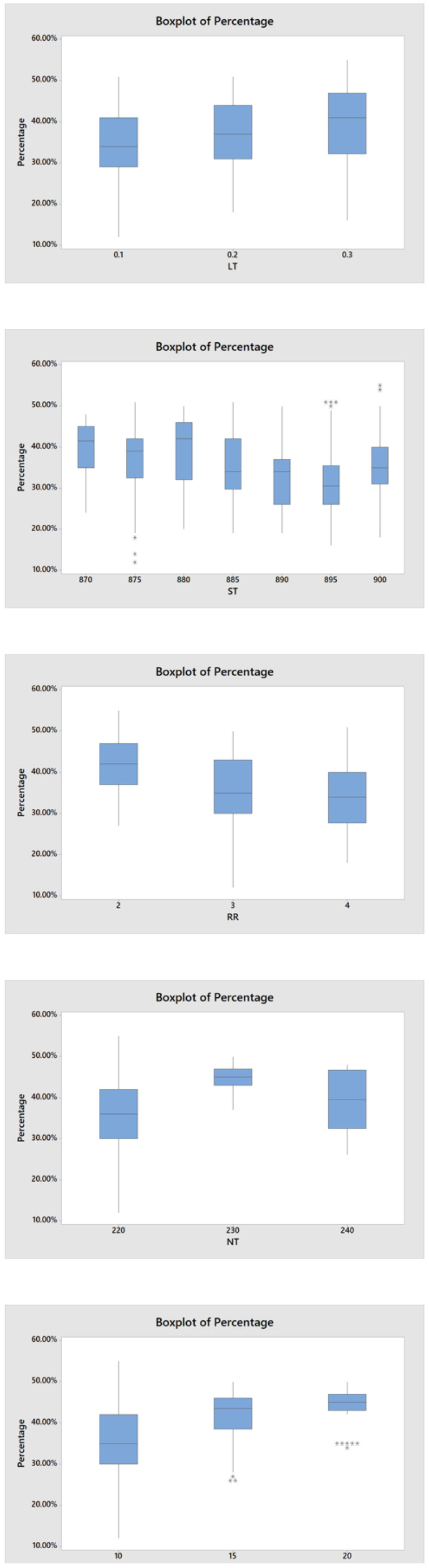
Boxplot of different manufacturing parameters on void percentage.

**Figure 9 materials-15-04292-f009:**
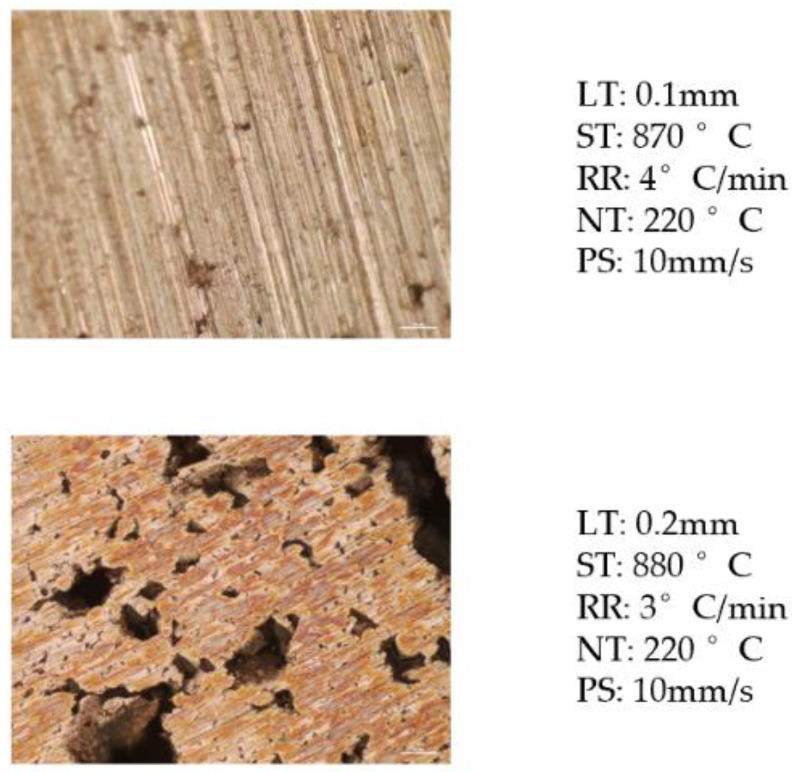
Interactions of manufacturing parameters.

**Table 1 materials-15-04292-t001:** Manufacturing Parameters.

Manufacturing Parameters	Values						
LT (mm)	0.1			0.2		0.3	
ST (°C)	870	875	880	885	890	895	900
RR (°C/min)	2			3		4	
NT (°C)	220			230		240	
PS (mm/s)	10			15		20	

**Table 2 materials-15-04292-t002:** Void percentage example.

Manufacturing Parameters					V (cm³)	M_Cal (g)	M (g)	Void Percentage
LT (mm)	ST (°C)	RR (°C/min)	NT (°C)	PS (mm/s)				
0.1	875	3	220	10	1.11	8.15	7.25	12%
0.2	870	4	240	15	1.02	5.82	3.93	48%

**Table 3 materials-15-04292-t003:** ANOVA analysis results.

Manufacturing Parameters	*p*-Value
LT (mm)	<2.2e-16
ST (°C)	<2.2e-16
NT (°C)	<2.2e-16
PS (mm/s)	8.330e-06
RR (°C/min)	<2.2e-16

**Table 4 materials-15-04292-t004:** MSE of ML algorithms.

ML Algorithms	MSE
SVR	0.0080165
NN	0.0022957

## Data Availability

Not Applicable.

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
