# Peer review of "Machine Learning-Based Void Percentage Analysis of Components Fabricated with the Low-Cost Metal Material Extrusion Process"

_materials, 2022, doi:10.3390/ma15124292_

Round 1

Reviewer 1 Report

This paper aims to apply machine learning algorithms to predict the size of the internal voids in parts fabricated by material extrusion. This research could be considered to accept after revising these flaws below:

  1. In abstract and introduction, the significance of research, current issues and the advance of applied method are not described clearly. The author should consider to rewrite the abstract and introduction.

  1. The microstructure (MS) seems not to be necessary to use abbreviation.

      3.SVR and NN are both popular machine learning algorithms. Is it necessary to compare the two methods?

  1. The processing of all figures are too rough and should be modified.

      5. The language of paper should be modified detailly.

Author Response

This paper aims to apply machine learning algorithms to predict the size of the internal voids in parts fabricated by material extrusion. This research could be considered to accept after revising these flaws below:

  1. In abstract and introduction, the significance of research, current issues and the advance of applied method are not described clearly. The author should consider to rewrite the abstract and introduction.

Thank you for this advice. The Abstract and Introduction have been edited.

  1. The microstructure (MS)seems not to be necessary to use abbreviation.

Thank you so much for this suggestion. All MS has been changed to microstructure.

  1. SVR and NN are both popular machine learning algorithms. Is it necessary to compare the two methods?

Thanks a lot for this question. We agree with you, SVR and NN are both popular ML algorithms. In this research, the researchers just would like to find the more accurate one, and the results shows that the MSE of NN is much smaller than SVR. So, the conclusion is that in this research, NN is more accurate than SVR.

  1. The processing of all figures are too rough and should be modified.

 Thank you so much, some figures have been changed.

  1. The language of paper should be modified detailly.

Thank you for this suggestion. Some sentences of the paper have been edited. See the revised manuscript.

Reviewer 2 Report

This paper presents Machine Learning-based Void Percentage Analysis of Components fabricated with the Low-cost Metal Material Extrusion Process.

The paper is interesting in general. However, most of the figures are not in presentable quality. These must be improved.

Comparative results for different machine learning models should be provided in detail. For example, refer to Composite Structures, 171 227–250

More numerical results should be added.

The  paper needs a significant revision before any further consideration.

Author Response

This paper presents Machine Learning-based Void Percentage Analysis of Components fabricated with the Low-cost Metal Material Extrusion Process.

The paper is interesting in general. However, most of the figures are not in presentable quality. These must be improved.

Figures were re-made.

Comparative results for different machine learning models should be provided in detail. For example, refer to Composite Structures, 171 227–250

Thanks a lot. Some more discussions on ML have been added to the paper.

More numerical results should be added.

Thank you so much. More discussions on results have been added in this paper.

The paper needs a significant revision before any further consideration.

Thanks a lot. A major edition has been done on this paper.

Reviewer 3 Report

The manuscript can not be accepted for publication for following reasons:

1.       The abstract needs to be improved, showing the key results of the study.

2.       The purpose of the paper should be clearly explained. Please, explain the main aim of the paper.

3.       The results are very poor described. The resutls are not discussed. There is no confrontation of the obtained research results with the literature data. Please, improve the results and add discussion. This section should be extensively widened.

Author Response

The manuscript cannot be accepted for publication for following reason.

  1. The abstract needs to be improved, showing the key results of the study.

Thank you for this suggestion. The abstract has been rewritten.

  1. The purpose of the paper should be clearly explained. Please, explain the main aim of the paper.

Thanks a lot. The following sentence has been added to explain the main aim of this paper: “the main aim is to improve the quality of the parts fabricated by LCMME with the help of ML.”

  1. The results are very poor described. The results are not discussed. There is no confrontation of the obtained research results with the literature data. Please, improve the results and add discussion. This section should be extensively widened.

Thank you so much. More discussions have been added in the results section.

Reviewer 4 Report

Comments: This article requires extensive corrections

1. Abstract is poorly written, it looks like introductory discussion. Please follow some good research articles. You can follow these articles about how they wrote. You can state which algorithm is best suited for this study in Abstract and the key results.

a. https://www.ncbi.nlm.nih.gov/pmc/articles/PMC8399339/

b. https://link.springer.com/article/10.1007/s40964-021-00192-4

2. Please follow these articles for writing a good introduction. 

a. https://link.springer.com/article/10.1007/s11665-021-05664-w

b.https://www.sciencedirect.com/science/article/abs/pii/S2352214321000605

c. https://www.astm.org/jte20170589.html

d. https://www.astm.org/mpc20180164.html

3. Figure 4 is a poor resolution. Please create a high quality figure.

4. Line 142, authors stated that compare with the void percentages of the above two examples, the quality of the first one is better than the second, since it has a smaller void percentage. Please discuss the reasons and justify with appropriate literature. It looks awkward if the author stated first one or second one. It is better author if the authors write Sample 1, sample 2 or experiment 1 or 2.

5. Please write the appropriateness of the ML algorithms like pros and cons and suitability.

6. Conclusion is poorly written, please mention the results, reasons, justifications very briefly, follow some good research work's conclusion.

Author Response

Comments: This article requires extensive corrections

  1. Abstract is poorly written, it looks like introductory discussion. Please follow some good research articles. You can follow these articles about how they wrote. You can state which algorithm is best suited for this study in Abstract and the key results. https://www.ncbi.nlm.nih.gov/pmc/articles/PMC8399339/
    https://link.springer.com/article/10.1007/s40964-021-00192-4

Thank you for this advice, the abstract has been rewritten.

  1. Please follow these articles for writing a good introduction.
    https://link.springer.com/article/10.1007/s11665-021-05664-w
    https://www.sciencedirect.com/science/article/abs/pii/S2352214321000605 
    https://www.astm.org/jte20170589.html
    https://www.astm.org/mpc20180164.html

Thank you so much, the introduction section has been edited.

  1. Figure 4 is a poor resolution. Please create a high-quality figure.

Thank you so much, Figure 4 has been changed.

  1. Line 142, authors stated that compare with the void percentages of the above two examples, the quality of the first one is better than the second, since it has a smaller void percentage. Please discuss the reasons and justify with appropriate literature. It looks awkward if the author stated first one or second one. It is better author if the authors write Sample 1, sample 2 or experiment 1 or 2.

Thank you for this advice, this sentence has been changed into “the first sample has a smaller void percentage than the second, which means that this sample is more solid and the mechanical property will be better [9], and thus, the quality of the first one is better than the second one.”

  1. Please write the appropriateness of the ML algorithms like pros and cons and suitability.

Thanks a lot. Some more discussions on ML have been added to the paper.

  1. Conclusion is poorly written, please mention the results, reasons, justifications very briefly, follow some good research work's conclusion.

Thank you so much for this suggestion. More discussions have been added to conclusion section.

Round 2

Reviewer 1 Report

 This reserch could be considered to accept in present form.

Author Response

Thank you so much for your decision.

Reviewer 2 Report

The comments given in the last review round have not been addressed carefully.

Author Response

Thank you so much for your suggestions. More detail discussion on ML has been added in section 3.3, and a new section, 4. Discussion, has been added into this paper. In this new section, more numerical results have been presented.

Reviewer 3 Report

Dear Authors,

There is still no discussion of the research results. There is no confrontation of the obtained research results with the publications of other authors. Research results should be widely discussed.

Best regards

Author Response

Thank you so much for your suggestions, a new section, 4. Discussion, has been added into this paper.

Reviewer 4 Report

Accept as it is

Author Response

Thank you so much for your decision.